# Neural Mutual Information Estimation with Vector Copulas

**Yanzhi Chen**[1], **Zijing Ou**[2], **Adrian Weller**[1,3], **Michael U. Gutmann**[4]
[1]University of Cambridge, [2]Imperial College London, [3]Alan Turing Institute, [4]University of Edinburgh

## Abstract

Estimating mutual information (MI) is a fundamental task in data science and machine learning. Existing estimators mainly rely on either highly flexible models (e.g., neural networks), which require large amounts of data, or overly simplified models (e.g., Gaussian copula), which fail to capture complex distributions. Drawing upon recent vector copula theory, we propose a principled interpolation between these two extremes to achieve a better trade-off between complexity and capacity. Experiments on state-of-the-art synthetic benchmarks and real-world data with diverse modalities demonstrate the advantages of the proposed estimator.

## 1 Introduction

Mutual information (MI) is a fundamental measure of the statistical dependence between random variables (RVs). Compared to other dependence measures, MI stands out due to its equitability and generality [1, 2]: it can capture non-linear dependence of any form and can handle RVs with any dimensionalities, rendering it a powerful measure for quantifying statistical dependence. In data science, MI is widely used to analyze the relationships between protein sequences [3] and gene profiles [4, 5], as well as to assess feature importance and redundancy [6]. In machine learning, MI broadly serves as a learning objective and regularizer [7, 8, 9, 10, 11, 12], with diverse applications to representation learning [7, 8, 9, 13, 14], generative modeling [10], fairness and privacy [15, 16], etc.

A wide range of powerful, neural MI estimators have been developed [17, 18, 19, 20, 21, 22, 23]. Most of these estimators rely on a *single, unconstrained* network to approximate certain quantities—such as the joint density $p(\mathbf{x}, \mathbf{y})$ or the density ratios $p(\mathbf{x}, \mathbf{y})/p(\mathbf{x})p(\mathbf{y})$—during MI estimation. While neural networks as universal functional approximators can, in theory, approximate arbitrary functions given sufficient data [24, 25], in practice we often only have a small set of data. Indeed, theoretical studies have shown that such *distribution-free* treatment of MI estimation will inevitably suffer from requiring an exponential sample size [26, 27, 28, 29]. A straightforward remedy is to restrict the model to simpler classes—for instance, assuming that the data is approximately Gaussian. However, these assumptions are often overly simplistic to capture complex distributions in reality.

Recent advances in vector copula theory [30] offer a promising avenue for addressing this dilemma. Vector copula theory extends classical copula theory [31] by generalizing it from *univariate* to *vector* marginals. It reveals that the multivariate marginals and the dependence structure (i.e., the vector copula) of a joint distribution are fully disentangled. This disentanglement motivates a more fine-grained way for making assumption in MI estimation, where we impose lightweight yet reasonable assumptions solely on the *vector copula* rather than on the *entire distribution*. Crucially, the complexity of the vector copula can be adaptively adjusted through efficient vector copula selection, allowing for an optimal trade-off between capacity and complexity. Experiments on state-of-the-art synthetic benchmarks and real-world data demonstrate the competitiveness of our estimator against state-of-the-art estimators. In summary, the main contributions of this work are three-fold:

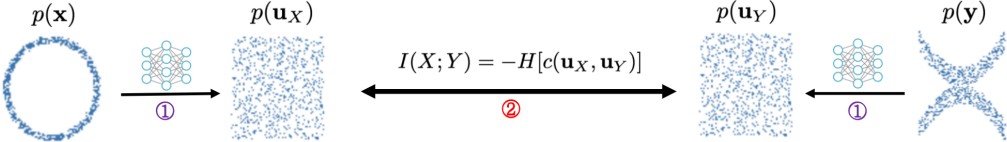

Figure 1: **Overview of the proposed vector copula-based MI estimator** (VCE), which explicitly disentangles the modeling of marginal distribution and dependence structure (i.e., the vector copula). VCE first respectively computes the vector ranks $\mathbf{u}_X$ and $\mathbf{u}_Y$ corresponding to the two marginal variables $X$ and $Y$ with flow models (①). It then finds the vector copula $c \in \mathcal{C}$ from the vector copula pool $\mathcal{C}$ that best matches with the joint distribution $p(\mathbf{u}_X, \mathbf{u}_Y)$ of the estimated vector ranks (②). Mutual information $I(X;Y)$ is computed as the negative differential entropy of the vector copula $c$, which itself is irrelevant to the two marginal distributions $p(\mathbf{x})$ and $p(\mathbf{y})$.

- We develop a divide-and-conquer MI estimator based on recent vector copula theory, which explicitly disentangles marginal distributions and dependence structure in MI estimation;
- We reinterpret existing estimators through the lens of vector copula, revealing that they correspond to varying parameterization and learning strategies of vector copula with various trade-offs;
- We provide consistency and error analysis of our estimator, along with extensive empirical evaluation on diverse test cases covering multiple modalities, marginal patterns and dependence structures.

Code containing both our method and state-of-the-art neural estimators is available in [github repo].

## 2 Preliminaries

Throughout this work, we use upper case letters (e.g. $X$) to denote random variables and lower case letters (e.g. $\mathbf{x}$) to denote their instances. We use $\mathcal{U}[0,1]^d$ or $\mu$ to denote the uniform distribution on $[0,1]^d$ and use $\mathcal{N}$ to denote Gaussian distribution on $\mathbb{R}^d$. $\nabla$ denotes the gradient and $J_{\mathbf{x}}\mathbf{y}$ denotes the Jacobian of $\mathbf{y}$ w.r.t $\mathbf{x}$. The symbol $\#$ denotes the push-forward operation.

### 2.1 Mutual information and its estimation

The mutual information (MI) between variables $X$ and $Y$ is defined as the Kullback-Leibler (KL) divergence between the joint distribution $p(\mathbf{x}, \mathbf{y})$ and the product of marginal distributions $p(\mathbf{x})p(\mathbf{y})$:

$$I(X;Y) = KL[p(\mathbf{x}, \mathbf{y})\|p(\mathbf{x})p(\mathbf{y})] = \mathbb{E}\left[\log \frac{p(\mathbf{x}, \mathbf{y})}{p(\mathbf{x})p(\mathbf{y})}\right] \tag{1}$$

In this work, we consider estimating $I(X;Y)$ from an empirical dataset $\mathcal{D} = \{\mathbf{x}^{(i)}, \mathbf{y}^{(i)}\}_{i=1}^n$. Several neural network-based methods have been developed for MI estimation:

**Generative estimators**. These methods leverage generative models to approximate the various distributions in (1) or their equivalents, and use the learned generative models to construct an MI estimate [17, 29, 18, 32, 19]. The accuracy of generative estimators crucially depends on the quality of the learned generative models. Simpler models (e.g. Gaussian copula) are easy to learn but may fail to adequately capture the true data distribution [33, 34]. In contrast, complex models (e.g. flow-based models [35, 36, 37, 38] and diffusion models [19]) offer greater expressiveness but can be challenging to optimize, in particular if the amount of data is insufficient or the data dimensionality is high.

**Discriminative estimators**. These methods train a neural network $f$ with samples $\mathbf{x}, \mathbf{y} \sim p(\mathbf{x}, \mathbf{y})$ and samples $\mathbf{x}, \mathbf{y} \sim p(\mathbf{x})p(\mathbf{y})$ to estimate the density ratio $p(\mathbf{x}, \mathbf{y})/p(\mathbf{x})p(\mathbf{y})$ [20, 21, 39, 40, 41, 32, 23]. Once trained, the learned density ratio can either be used in (1) or in the Donsker-Varadhan (DV) representation [42] to obtain an MI estimate. Discriminative methods avoid directly modeling densities, however they are prone to the curse of *high-discrepancy* [40, 41, 29, 32], which occurs if $p(\mathbf{x}, \mathbf{y})$ and $p(\mathbf{x})p(\mathbf{y})$ differ significantly — for instances, cases with high MI or high-dimensional data. Several advanced methods were proposed to alleviate this issue, including clipping the network outputs [32], introducing reference distributions [41], avoiding computing the partition function [23].

## 2.2 Vector copula

**Vector copula theory**. The recent vector copula theory [30] provides a principled framework for modeling and analyzing the dependence between *multivariate* random variables. It extends classical copula theory [31] by considering 'vector' marginals. We begin by the concept of vector ranks:

**Definition 1** (Vector rank). *Let $p$ be an absolutely continuous distribution on $\mathbb{R}^d$ with support in a convex set. Let $\mu$ be the uniform distribution on $[0,1]^d$. There exists a convex function $\psi$ such that $\nabla\psi\#\mu = p$ and $\nabla\psi^{-1}\#p = \mu$. $\mathbf{u} := \nabla\psi^{-1}$ is called the vector rank associated with $p$ [43].*

When $d = 1$, vector rank reduces to standard scalar rank. Intuitively, vector rank transforms a multivariate distribution $p$ to a (multivariate) uniform distribution $\mu$, entirely removing its characteristics.

In the text below, we slightly overload this definition and use the term 'vector rank' to refer to both the vector rank function $\mathbf{u}(\cdot)$ and also the corresponding random variable $\mathbf{u}$ induced by this function.

**Definition 2** (Vector copula). *Let $\mathbf{u}_X$ and $\mathbf{u}_Y$ be the vector ranks corresponding to $p(\mathbf{x})$ and $p(\mathbf{y})$ respectively. A vector copula $C(\mathbf{u}_X, \mathbf{u}_Y)$ is a cumulative distribution function on $[0,1]^{d_X+d_Y}$ with uniform marginals on $C(\mathbf{u}_X) = \mathcal{U}[0,1]^{d_X}$ and $C(\mathbf{u}_Y) = \mathcal{U}[0,1]^{d_Y}$. The probabilistic density function corresponding to $C$ is called* vector copula density *and is denoted as $c(\mathbf{u}_X, \mathbf{u}_Y)$ [30].*

Given the above definition, we have the following result [30] generalizing the Sklar theorem [31].

**Theorem 1** (Vector Sklar Theorem). *Let $X \in \mathbb{R}^{d_X}$ and $Y \in \mathbb{R}^{d_Y}$ be two random variables with joint distribution $p(\mathbf{x}, \mathbf{y})$ on $\mathbb{R}^{d_X+d_Y}$. For any absolutely continuous distributions $p(\mathbf{x}, \mathbf{y})$ with support in a convex set, there exist an unique function $c(\cdot, \cdot)$, such that*

$$p(\mathbf{x}, \mathbf{y}) = p(\mathbf{x})p(\mathbf{y})c(\mathbf{u}_X, \mathbf{u}_Y) \tag{2}$$

*where $\mathbf{u}_X$ and $\mathbf{u}_Y$ are the vector ranks computed for $\mathbf{x}$ and $\mathbf{y}$ respectively. The function $c$ equals to the vector copula density associated with $\mathbf{u}_X$ and $\mathbf{u}_Y$ [30].*

The vector Sklar theorem suggests that for a distribution $p(\mathbf{x}, \mathbf{y})$, its marginal distributions and the joint dependence structure are entirely disentangled, with the latter fully characterized by the vector copula density $c$. Note that here we focus on the case of two RVs; we refer to [30] for general cases.

**Instances of vector copula**. We discuss several instances of vector copula related to our work. One important instance is the *vector Gaussian copula* [30]. This model assumes that the joint dependence structure admits a Gaussian structure, with its vector copula $C^{\mathcal{N}}$ being

$$C^{\mathcal{N}}(\mathbf{u}_X, \mathbf{u}_Y) = \Phi(\phi^{-1}(\mathbf{u}_X), \phi^{-1}(\mathbf{u}_Y); \mathbf{0}, \Sigma) \tag{3}$$

where $\Sigma = [[\mathbf{I}_X, \Sigma_{XY}], [\Sigma_{XY}^\top, \mathbf{I}_Y]]$ is a p.s.d matrix whose blocks $\mathbf{I}_X \in \mathbb{R}^{d_X \times d_X}$ and $\mathbf{I}_Y \in \mathbb{R}^{d_Y \times d_Y}$ are identity matrices. $\Phi(\cdot)$ is the cumulative distribution function of multivariate normal distribution and $\phi(\cdot)$ is the (element-wise) cumulative distribution function of univariate normal distribution. Equivalently, a vector Gaussian copula can be defined by its data generation process: $\epsilon \sim \mathcal{N}(\epsilon; \mathbf{0}, \Sigma)$, $\mathbf{u}_X = \phi(\epsilon_{\leq d_X})$, $\mathbf{u}_Y = \phi(\epsilon_{> d_X})$, with $\epsilon_{\leq d_X}$ and $\epsilon_{> d_X}$ being the first $d_X$ and the remaining dimensions of $\epsilon$ respectively. An analytic expression for $c^{\mathcal{N}}$ can be derived accordingly.

Other useful instances of vector copula include $t$-vector copula, Archimedean vector copula and Kendall vector copula, which correspond to different inductive biases about the dependence structure.

## 3 Methodology

In this section, we propose a new mutual information (MI) estimator based on vector copula theory. The core of our method is Theorem 2, which establishes a connection between MI and vector copula:

**Theorem 2** (MI is vector copula entropy). *The mutual information $I(X; Y)$ is the negative differential entropy of the vector copula density:*

$$I(X; Y) = -H[c(\mathbf{u}_X, \mathbf{u}_Y)] \tag{4}$$

*where $\mathbf{u}_X$ and $\mathbf{u}_Y$ are the vector ranks corresponding to $p(\mathbf{x})$ and $p(\mathbf{y})$ respectively.*

*Proof*: Please refer to Appendix A. □

| **Algorithm 1** Vector copula MI estimate (VCE) | **Algorithm 2** Vector copula MI estimate' (VCE') |
|---|---|
| **Input:** data $\mathcal{D} = \{\mathbf{x}^{(i)}, \mathbf{y}^{(i)}\}_{i=1}^n$ | **Input:** data $\mathcal{D} = \{\mathbf{x}^{(i)}, \mathbf{y}^{(i)}\}_{i=1}^n$ |
| **Output:** estimated $\hat{I}(X;Y)$ | **Output:** estimated $\hat{I}(X;Y)$ |
| **Parameters:** flows $f_X, f_Y$, copulas $\{c_1, ..c_M\}$ | **Parameters:** flows $f_X, f_Y$, ratio estimator $r$ |
| **Initialization:** $\mathcal{D} = \mathcal{D}_{train} \cup \mathcal{D}_{val}$, $K = 1$, | **Initialization:** reference copula $c'$, $\mathcal{D}' = \emptyset$ |
| ▷ *Marginal distributions learning* | ▷ *Marginal distributions learning* |
| learn $f_X$ with $\mathcal{D}_X = \{\mathbf{x}^{(i)}\}_{i=1}^n$ by FM; | learn $f_X$ with $\mathcal{D}_X = \{\mathbf{x}^{(i)}\}_{i=1}^n$ by FM; |
| learn $f_Y$ with $\mathcal{D}_Y = \{\mathbf{y}^{(i)}\}_{i=1}^n$ by FM; | learn $f_Y$ with $\mathcal{D}_Y = \{\mathbf{y}^{(i)}\}_{i=1}^n$ by FM; |
| **for** $i$ in 1 to $n$ **do** | **for** $i$ in 1 to $n$ **do** |
| $\quad$ compute $\hat{\mathbf{u}}_X^{(i)} = \texttt{rank}(f_X(\mathbf{x}^{(i)}))$; | $\quad$ compute $\hat{\mathbf{u}}_X^{(i)} = \texttt{rank}(f_X(\mathbf{x}^{(i)}))$; |
| $\quad$ compute $\hat{\mathbf{u}}_Y^{(i)} = \texttt{rank}(f_Y(\mathbf{y}^{(i)}))$; | $\quad$ compute $\hat{\mathbf{u}}_Y^{(i)} = \texttt{rank}(f_Y(\mathbf{y}^{(i)}))$; |
| **end for** | **end for** |
| ▷ *Vector copula density estimation* | ▷ *Vector copula density estimation* |
| **repeat** | **repeat** |
| $\quad$ set $c(\mathbf{u}_X, \mathbf{u}_Y) = \frac{1}{K}\sum_{k=1}^K p_k c_k(\mathbf{u}_X, \mathbf{u}_Y)$; | $\quad$ sample $\mathbf{u}_X^{(j)}, \mathbf{u}_Y^{(j)} \sim c'(\mathbf{u}_X, \mathbf{u}_Y)$; |
| $\quad$ $\hat{c} = \arg\max_c \mathbb{E}_{\hat{\mathbf{u}}_X, \hat{\mathbf{u}}_Y \sim \mathcal{D}_{train}}[\log c(\hat{\mathbf{u}}_X, \hat{\mathbf{u}}_Y)]$; | $\quad$ $\mathcal{D}' \leftarrow \mathcal{D}' \cup \{\mathbf{u}_X^{(j)}, \mathbf{u}_Y^{(j)}\}$; |
| $\quad$ $\mathcal{L}_{val} \leftarrow \mathbb{E}_{\hat{\mathbf{u}}_X, \hat{\mathbf{u}}_Y \sim \mathcal{D}_{val}}[\log c(\hat{\mathbf{u}}_X, \hat{\mathbf{u}}_Y)]$; | **until** $|\mathcal{D}'| = n$ |
| $\quad$ $K \leftarrow 2K$; | train $r$ to classify samples from $\mathcal{D}$ and $\mathcal{D}'$; |
| **until** no improvement on $\mathcal{L}_{val}$ | set $\hat{c}(\mathbf{u}_X, \mathbf{u}_Y) = r(\mathbf{u}_X, \mathbf{u}_Y) \cdot c'(\mathbf{u}_X, \mathbf{u}_Y)$; |
| **return** $\hat{I}(X;Y) = \frac{1}{n}\sum_{i=1}^n \log \hat{c}(\hat{\mathbf{u}}_X^{(i)}, \hat{\mathbf{u}}_Y^{(i)})$ | **return** $\hat{I}(X;Y) = \frac{1}{n}\sum_{i=1}^n \log \hat{c}(\hat{\mathbf{u}}_X^{(i)}, \hat{\mathbf{u}}_Y^{(i)})$ |

This theorem generalizes the results of [44, 45] from univariate to vector marginals[1]. It establishes that MI depends solely on the vector copula, which itself is invariant to marginal distributions. Notably, the theorem also reveals that the pointwise mutual information (PMI) i.e. $p(\mathbf{x}, \mathbf{y})/p(\mathbf{x})p(\mathbf{y})$ can equivalently be viewed as a *density* $c(\mathbf{u}_X, \mathbf{u}_Y)$ in its own right, in contrast to the vast majority of existing works [46, 47, 48, 49] which continue to treat PMI as a *density ratio*. This shift in perspectives opens us new possibility in the parameterization and learning of the PMI, including directly modeling it as a normalized density learned via MLE, as will be discussed later.

Theorem 2 immediately suggests a new *divide-and-conquer* approach for MI estimation: we can first estimate the vector ranks $\mathbf{u}_X$ and $\mathbf{u}_Y$, followed by subsequent learning of the vector copula $c$[2]:

$$I(X;Y) \approx \hat{I}(X;Y) \coloneqq \frac{1}{n}\sum_{i=1}^n \log \hat{c}(\hat{\mathbf{u}}_X^{(i)}, \hat{\mathbf{u}}_Y^{(i)}) \tag{5}$$

where $\hat{\mathbf{u}}_X$, $\hat{\mathbf{u}}_Y$ and $\hat{c}$ are the empirical estimates to $\mathbf{u}_X$, $\mathbf{u}_Y$ and $c$ respectively.

We discuss below several potential advantages of the above divide-and-conquer estimation strategy:

- By disentangling the *modeling* of marginals distribution and copula, we can use differently-sized models in their parameterization, avoiding using a single overly flexible or overly simplified model for the entire distribution. This leads to a better trade-off between model complexity and capacity;

- By disentangling the *learning* of marginals and copula, we can reuse the pre-trained marginals across multiple copula choices with varying complexities, allowing model selection to be performed solely in the copula space in a computational efficient way. It also reduces overall learning difficulty.

In the following, we elaborate methods to estimate the vector ranks and the vector copula respectively.

---

[1]Building upon classic copula, the theory in [44, 45] only holds for bivariate cases, and generalizing their results to high-dimensional cases require non-trivial formulation and derivation—precisely our key contribution.

[2]Alternatively, one may also learn the marginals $\hat{p}(\mathbf{x}), \hat{p}(\mathbf{y})$ and the vector copula $\hat{c}$ jointly. However, joint learning can be ill-posed [50]. Our ablation study in Appendix B2 suggests that separate learning is more robust.

## 3.1 Marginal distribution learning

In this step, we learn the two marginal distributions $p(\mathbf{x})$ and $p(\mathbf{y})$ with flexible flow-based models [35, 36, 37, 38] and use them to compute the vector ranks $\mathbf{u}_X$ and $\mathbf{u}_Y$.

**Flow-based modeling of marginals**. Let $f_X : \mathbb{R}^{d_X} \to \mathbb{R}^{d_X}$ and $f_Y : \mathbb{R}^{d_Y} \to \mathbb{R}^{d_Y}$ be two flow-based models and let $p_{f_X}(\mathbf{x})$ and $p_{f_Y}(\mathbf{y})$ be the densities induced by $f_X$ and $f_Y$ respectively. We respectively learn $f_X$ and $f_Y$ with data $\mathbf{x} \sim p(\mathbf{x})$ and data $\mathbf{y} \sim p(\mathbf{y})$ by flow matching [38]:

$$\min_{f_X} \mathbb{E}[\mathcal{L}_{\text{FM}}(\mathbf{x}; f_X)], \qquad \min_{f_Y} \mathbb{E}[\mathcal{L}_{\text{FM}}(\mathbf{y}; f_Y)] \tag{6}$$

where $\mathcal{L}_{\text{FM}}$ denotes the flow-matching loss [38]. Upon convergence, $f_X$ and $f_Y$ respectively transform the two marginals to a standard normal distribution: $\mathcal{N}(\mathbf{0}, \mathbf{I}) \approx f_X \# p(\mathbf{x})$ and $\mathcal{N}(\mathbf{0}, \mathbf{I}) \approx f_Y \# p(\mathbf{y})$.

**Vector ranks computation**. With the learned flows $f_X$ and $f_Y$, we compute the vector ranks as:

$$\hat{\mathbf{u}}_X^{(i)} = \texttt{rank}(f_X(\mathbf{x}^{(i)})), \qquad \hat{\mathbf{u}}_Y^{(i)} = \texttt{rank}(f_Y(\mathbf{y}^{(i)})) \tag{7}$$

where $\texttt{rank}_d(\boldsymbol{\epsilon}) = \frac{1}{n+1}\sum_{j=1}^{n} \mathbf{1}[\epsilon_d \geq \epsilon_d^{(j)}]$ is the element-wise ranking function that computes the scalar ranks for each of the dimension in $\epsilon$. Given universal density approximators $f_X$, $f_Y$, $\hat{\mathbf{u}}_X$ and $\hat{\mathbf{u}}_Y$ serve as consistent estimates of the true vector ranks $\mathbf{u}_X$ and $\mathbf{u}_Y$.

While the joint density $p(\mathbf{x}, \mathbf{y})$ is often challenging to estimate, the marginal distributions $p(\mathbf{x})$ and $p(\mathbf{y})$ are typically far easier to learn due to their lower dimensionality. It is thus reasonable to expect that $\hat{\mathbf{u}}_X$ and $\hat{\mathbf{u}}_Y$ are close approximations to $\mathbf{u}_X$ and $\mathbf{u}_Y$ in moderate dimensionality settings.

*Remark*. The above process of estimating $\mathbf{u}_X$ and $\mathbf{u}_Y$ can be viewed as a generalization of classic copula transformation in MI estimation, where we compute vector ranks rather than scalar ranks.

## 3.2 Vector copula estimation

In this step, we learn the vector copula $c$ with the previously estimated vector ranks $\hat{\mathbf{u}}_X$ and $\hat{\mathbf{u}}_Y$, leveraging a model-based parameterization and a careful model selection strategy.

**Model-based parameterization of copula**. As noted earlier, any parametric model can be used to represent the vector copula $c$, regardless of whether an analytical PMI is available. In this work, we parameterize $c$ as a mixture of existing parametric vector copulas [30] from the copula pool, whose model complexity can be well controlled by tuning the number of mixture components:

$$c(\mathbf{u}_X, \mathbf{u}_Y) = \sum_{k=1}^{K} p_k c_k(\mathbf{u}_X, \mathbf{u}_Y), \tag{8}$$

where $\sum_{k=1}^{K} p_k = 1$ and each $c_k \in \mathcal{C}$ is selected from the predefined pool $\mathcal{C}$ of vector copulas. Any inductive bias about the dependence structure can be used to guide copula selection. Here, we simply implement each $c_k$ as a vector Gaussian copula and learn $c$ by maximum likelihood estimate (MLE):

$$\max_{c} \mathbb{E}[\log c(\mathbf{u}_X, \mathbf{u}_Y)] \tag{9}$$

In theoretical analysis, we analyze why this copula design is a cheap yet reasonable modeling of $c$.

**Efficient model selection**. A key design in our method is the explicit exploration of the capacity–complexity trade-off in copula modeling, which is governed by the number of mixture components $K$. Here, we determine $K$ by cross validation, using negative log-likelihood (NLL) as the criterion. This process is computationally cheap: each copula is already lightweight, involving no neural networks; furthermore, different copulas can be trained in parallel using one single loss.

Algorithm 1 summarizes the main pipeline of the proposed vector copula-based estimator (VCE).

*Remark*. As an alternative to the above model-based parameterization, one may also adopt a reference-based parameterization for the vector copula, inspired by the design in [51]. Specifically, let $c'$ be a reference vector copula that is easy to sample (e.g. a vector Gaussian copula). We can learn $c$ by first estimating the density ratio $r = c/c'$ using samples from $c$ and $c'$ [52, 41, 39], then recover the vector copula $c$ as $c = r \cdot c'$; see Algorithm 2. By parameterizing $r$ as a deep neural network, this method allows for a more flexible modeling of $c$, at the cost of a less fine-grained control over its complexity.

# 4 Theoretical analysis

In this section, we analyze several important theoretical properties of the proposed VCE estimator.

**Proposition 1** (Consistency of VCE). *Assuming that (a) $f_X$ and $f_Y$ are universal PDF approximators with continuous support and (b) the number of mixture components $K$ in (8) is sufficiently large. Define $\hat{I}_n(X;Y) := \frac{1}{n} \sum_{i=1}^{n} \log \hat{c}(\hat{\mathbf{u}}_X^{(i)}, \hat{\mathbf{u}}_Y^{(i)})$. For every $\epsilon > 0$, there exists $n(\varepsilon) \in \mathbb{N}$, such that*

$$\left| \hat{I}_n(X;Y) - I(X;Y) \right| < \varepsilon, \quad \forall n \geq n(\varepsilon), a.s.$$

*Proof.* Please refer to Appendix A ∎

Additionally, we have the following result analyzing the estimation error w.r.t the quality of the learned marginals $p_{f_X}(\mathbf{x}), p_{f_Y}(\mathbf{y})$ and the estimated vector copula density $\hat{c}$.

**Proposition 2** (Error of vector copula-based MI estimate). *Let $\hat{\mathbf{u}}_X$ and $\hat{\mathbf{u}}_Y$ be the estimated vector ranks. Let $c(\hat{\mathbf{u}}_X, \hat{\mathbf{u}}_Y)$ be the true joint distribution of $\hat{\mathbf{u}}_X$ and $\hat{\mathbf{u}}_Y$, and $\hat{c}(\hat{\mathbf{u}}_X, \hat{\mathbf{u}}_Y)$ its estimate. Assuming that sufficient Monte Carlo samples are used to compute $\hat{I}(X;Y)$ in (5), we have*

$$\left| I(X;Y) - \hat{I}(X;Y) \right| \leq \left| H(\hat{\mathbf{u}}_X) + H(\hat{\mathbf{u}}_Y) \right| + KL[c(\hat{\mathbf{u}}_X, \hat{\mathbf{u}}_Y)) \| \hat{c}(\hat{\mathbf{u}}_X, \hat{\mathbf{u}}_Y)]$$

*where the first term on the RHS vanishes as $p_{f_X}(\mathbf{x}) \to p(\mathbf{x})$ and $p_{f_Y}(\mathbf{y}) \to p(\mathbf{y})$. In the limit of perfectly learned marginals, the error simplifies to*

$$|I(X;Y) - \hat{I}(X;Y)| = KL[c \| \hat{c}],$$

*with $c$ and $\hat{c}$ being the true vector copula density and estimated vector copula density, respectively.*

*Proof.* Please refer to Appendix A. ∎

Proposition 2 decomposes the estimation error of the proposed VCE estimator into two components:

- *Marginal estimation error.* Imperfect marginal estimations introduce a bias given by $|H(\hat{\mathbf{u}}_X) + H(\hat{\mathbf{u}}_Y)| > 0$, which diminishes as both marginals are learned more accurately (recall that ideally, we have $\hat{\mathbf{u}}_X \sim \mathcal{U}[0,1]^{d_X}$ and $\hat{\mathbf{u}}_Y \sim \mathcal{U}[0,1]^{d_Y}$). For data with moderate dimensionality, we expect this bias to be small, as the two marginals are with low-dimensionality, being easy to estimate.

- *Dependence structure modeling error.* This error arises from the discrepancy between the estimated copula $\hat{c}$ and the true copula $c$. It depends on two factors: (a) *capacity* - whether the parameterization of $\hat{c}$ is sufficiently expressive to approximate $c$; and (b) *complexity* - how easy $\hat{c}$ can be learned from the limited data. These factors highlight the importance of model selection for the copula $c$.

**Proposition 3** (Vector Gaussian copula as second-order approximation). *A vector Gaussian copula $c^{\mathcal{N}}$ corresponds to the second-order Taylor expansion of the true vector copula $c^*$ up to variable transformation.*

*Proof.* Please refer to Appendix A. ∎

This result explains our choice of using a mixture of Gaussian copulas as a cheap yet principled approximation to the true vector copula. A single vector Gaussian copula already offers a reasonable approximation of the true copula by capturing dependencies up to second order; higher-order interactions, if necessary, can be modeled by adding mixture components in a fully controllable way.

Finally, we have the following result regarding cases with weakly dependent random variables (RVs).

**Proposition 4** (Vector copula of independent RVs). *The vector copula corresponding to the product of marginals $p'(\mathbf{x}, \mathbf{y}) = p(\mathbf{x})p(\mathbf{y})$ is a vector Gaussian copula if $p'(\mathbf{x}, \mathbf{y})$ is absolutely continuous.*

*Proof.* Please refer to Appendix A. ∎

Proposition 4 suggests that if the two RVs $X$ and $Y$ are nearly independent, our estimator is *likely* to provide an accurate estimation of $I(X;Y)$ as the true vector copula is Gaussian-like, being close to the family of our copula design (8). For weakly dependent RVs, it is reasonable to expect that $p(\mathbf{x}, \mathbf{y})$ resembles a vector Gaussian copula, with the difference captured by the additional components in (8).

# 5    Reinterpreting existing MI estimators

In this section, we reinterpret existing MI estimators through the lens of vector copula theory, showing that they correspond to different parameterizations and learning strategies of the vector copula.

**Reinterpreting discriminative estimators**. Existing critic-based approach to MI estimation [20, 53, 39, 41, 32, 23] can be interpreted as parameterizing the vector copula $c(\mathbf{u}_X, \mathbf{u}_Y)$ using a feedforwarding neural network $f$:

$$c(\mathbf{u}_X, \mathbf{u}_Y) \propto e^{f(\mathbf{x}, \mathbf{y})} \tag{10}$$

which is learned by discerning samples from the joint $p(\mathbf{x}, \mathbf{y})$ and the product of marginals $p(\mathbf{x})p(\mathbf{y})$ (via e.g contrastive learning). Specifically, recall that the optimal critic $f$ in these methods corresponds to the log density ratio up to an additive constant $C$ [54]: $f(\mathbf{x}, \mathbf{y}) = \log p(\mathbf{x}, \mathbf{y})/p(\mathbf{x})p(\mathbf{y}) + C$, with the PMI itself equal to the vector copula density, as established by the vector Skalar theorem.

Compared to our model-based parameterization of the vector copula density in (8), this neural network parameterization is more flexible and can potentially capture more complex dependence structures. However, as discussed earlier, such distribution-free parameterizations lack complexity control, which may lead to a poor bias–variance trade-off. Furthermore, discriminative methods learns the vector copula by comparing distributions, which can be challenging if they differ significantly (e.g., in high-MI cases, see [40, 41, 29, 32]. In contrast, our main method learns the vector copula by maximum likelihood estimate (MLE), which is the most efficient consistent estimator for the copula.

**Reinterpreting generative estimators**. Many generative estimators for MI [17, 29, 18, 48, 32] require either learning the joint distribution $p(\mathbf{x}, \mathbf{y}) = p(\mathbf{x})p(\mathbf{y})c(\mathbf{u}_X, \mathbf{u}_Y)$ or the conditional distribution $p(\mathbf{y}|\mathbf{x}) = p(\mathbf{y})c(\mathbf{u}_X, \mathbf{u}_Y)$ using a single model. This process can be interpreted as learning the marginal distribution(s) and the vector copula simultaneously, with the two components parameterized *jointly* via a *single* generative model. Our method, on the contrary, explicitly separates the modeling and the learning of the marginal distributions $p(\mathbf{x}), p(\mathbf{y})$ from that of the vector copula $c(\mathbf{u}_X, \mathbf{u}_Y)$. This strategy not only enables a more fine-grained control over model complexity, but also mitigates the challenge of jointly learning the marginal distribution and the dependence structure—a strategy aligned with the spirit of classical copula transformations [47, 55, 56, 57] to simplify MI estimation.

We further discuss two recent works [48, 17] closely related to our work. These methods operate by respectively transforming the two RVs $X$ and $Y$ by two flow-based models, such that the joint distribution of the transformed data can be approximated by a distribution with an easy-to-compute MI (for instance, a Gaussian distribution). Their practical methods, $\mathcal{N}$-MIENF and DINE-Gaussian, can be reinterpreted as assuming the dependence structure as a vector Gaussian copula (see Lemma 3 in Appendix A4 for a detailed derivation):

$$c(\mathbf{u}_X, \mathbf{u}_Y) \approx c^{\mathcal{N}}(\mathbf{u}_X, \mathbf{u}_Y; \Sigma) \tag{11}$$

which corresponds to the case $K = 1$ in the VCE estimator and is accurate (only) if the true dependence is Gaussian-like. The possibility of using non-Gaussian base distribution is also discussed in [48], albeit without practical implementation. Additionally, the marginals and the vector copula in their method are learned jointly rather than separately as in our method, and they continue to treat PMI as a density ratio $p(\mathbf{x}, \mathbf{y})/p(\mathbf{x})p(\mathbf{y})$, unlike our method which treats it as a density $c(\mathbf{u}_X, \mathbf{u}_Y)$.

# 6    Experiments

**Baselines**. We consider five representative neural estimators in the field: MINE [20], InfoNCE [21], MRE [41], MINDE [19] and $\mathcal{N}$-MIENF [48]. The first three methods are critic-based whereas the latter three are generative model-based. MRE is chosen as the representative of state-of-the-art discriminative methods, which is specifically designed to address the high-discrepancy issue in these methods. MINDE is chosen to represent the state-of-the-art generative methods, which leverages powerful diffusion model in MI estimation. Further baselines are considered in Appendix B2.

**Hyperparams**. For the vector copula in VCE, we consider mixtures with $1, 4, 8, 16, 32$ components.

**Neural architecture, optimizer and training details**. Please refer to appendix B1 for more details.

In the following evaluation, we primarily focus on evaluating the VCE estimator (Algorithm 1), and present the results of the alternative VCE' estimator (Algorithm 2) in the appendix. All results are collected through 8 independent runs. Error bars reported are the standard deviations (std) of the runs.

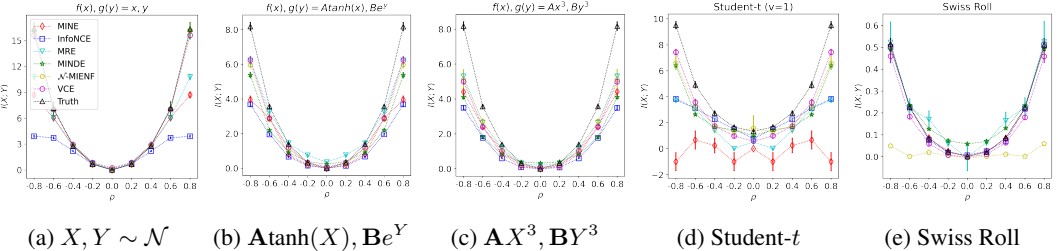

(a) $X, Y \sim \mathcal{N}$    (b) $\mathbf{A}\tanh(X), \mathbf{B}e^Y$    (c) $\mathbf{A}X^3, \mathbf{B}Y^3$    (d) Student-$t$    (e) Swiss Roll

Figure 2: Comparing MI estimators under various dependence strengths $\rho$. Data in cases (b)(c) are generated by first sampling $X, Y \sim \mathcal{N}$ as in case (a), then transforming them with the shown transformations. The dimensionalities of the data in the five cases are 64, 32, 32, 32, 2 respectively.

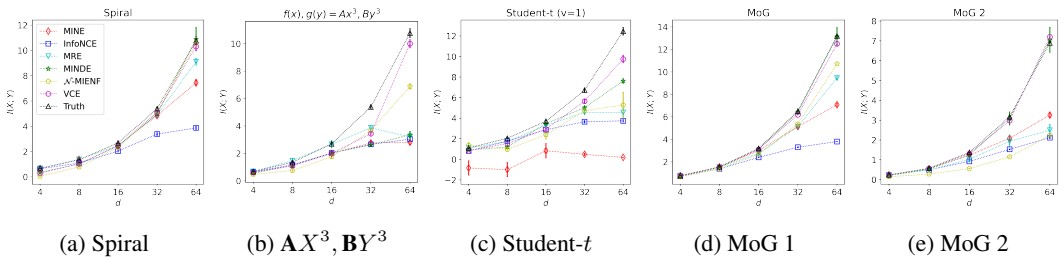

(a) Spiral    (b) $\mathbf{A}X^3, \mathbf{B}Y^3$    (c) Student-$t$    (d) MoG 1    (e) MoG 2

Figure 3: Comparing different MI estimators under various data dimensionality $d$ and fixed dependence levels. MoG corresponds to mixture of Gaussians. Spiral corresponds to spiral transformation.

## 6.1 Synthetic distributions

**Setups**. In [58], a diverse set of models with known MI are developed to comprehensively evaluate MI estimators. We consider representative cases from this benchmark, further extending it by (a) considering varying dependence strengths for each chosen case; (b) employing mixing matrices $\mathbf{A}, \mathbf{B}$ to couple the dimensions in $X$ and $Y$ respectively. We also include the mixture models in [49] to enrich our tests. Together, our test cases cover non-Gaussianity, skewness, heterogeneous marginals, long tails, low-dimensional manifold structure, coupling dimensions, high-dimensionality, varying dependence strengths and non-Gaussian dependence structure. Each test case contains $n = 10^4$ data.

**Results**. Figure 2 and Figure 3 compare the performance of different MI estimators[3]. Overall, VCE provides good estimates in *all* scenarios, consistently ranking among the top performers.

Compared to discriminative methods e.g., MINE and InfoNCE, VCE demonstrates significant advantages, particularly in high MI settings (e.g. strong dependence level $\rho$ or high dimensionality $d$). This advantage may be because our method avoids directly comparing two highly distinct distributions as in these methods, which is challenging. The advantage may also attribute to the better complexity-capacity trade-off in our method, which avoids an overly powerful model for the copula.

Compared to the generative method $\mathcal{N}$-MIENF, VCE demonstrates advantages in scenarios involving non-Gaussian dependence structures (see e.g. the MoG cases and 64D $t$-distribution). In such cases, $\mathcal{N}$-MIENF's assumption of a Gaussian dependence structure falls short in capturing the true dependence structure. This underscores the pitfalls of using a overly simplified model for the copula.

We specifically discuss two challenging cases highlighted in prior works [58, 19]: (a) Spiral transformation, which highly transforms the original data; and (b) multivariate $t$-distribution with degree of freedom $\nu = 1$, which exhibit heavy-tailed dependence. For these two highly challenging scenarios, VCE and MINDE are the only two methods that can simultaneously provide reasonable estimates in *both* cases, with VCE outperforming MINDE in other settings (see e.g., Figure 2.c and Figure 3.b).

---

[3]Comparison to classic copula-based MI estimators and further discriminative estimators is in Appendix B2.

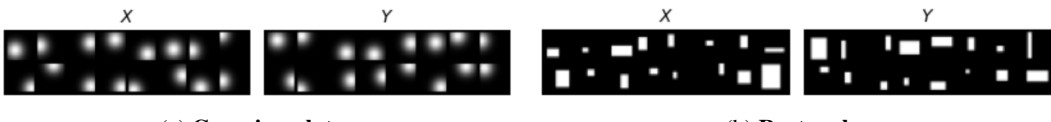

(a) **Gaussian plates**              (b) **Rectangles**

Figure 4: The image dataset [59], which contains images of rectangles and Gaussian plates.

| Method | Gaussian Plates | | | Rectangles | | |
|---|---|---|---|---|---|---|
| | $I(X;Y) = 1$ | $I(X;Y) = 3$ | $I(X;Y) = 7$ | $I(X;Y) = 1$ | $I(X;Y) = 3$ | $I(X;Y) = 7$ |
| MINE | $0.89 \pm 0.07$ | $2.86 \pm 0.24$ | $5.46 \pm 0.27$ | $0.81 \pm 0.13$ | $\mathbf{2.57 \pm 0.26}$ | $5.39 \pm 0.23$ |
| InfoNCE | $0.86 \pm 0.14$ | $2.63 \pm 0.13$ | $3.83 \pm 0.12$ | $0.78 \pm 0.17$ | $2.49 \pm 0.28$ | $3.86 \pm 0.15$ |
| MRE | $1.23 \pm 0.16$ | $2.85 \pm 0.21$ | $5.91 \pm 0.28$ | $0.82 \pm 0.24$ | $2.56 \pm 0.48$ | $\mathbf{5.45 \pm 0.31}$ |
| $\mathcal{N}$-MIENF | $0.74 \pm 0.12$ | $2.42 \pm 0.16$ | $3.85 \pm 0.22$ | $0.54 \pm 0.13$ | $0.76 \pm 0.14$ | $1.54 \pm 0.11$ |
| VCE | $\mathbf{0.92 \pm 0.04}$ | $\mathbf{2.93 \pm 0.12}$ | $\mathbf{6.53 \pm 0.36}$ | $\mathbf{0.83 \pm 0.12}$ | $2.27 \pm 0.23$ | $5.02 \pm 0.14$ |

Table 1: Comparing different MI estimators on the image benchmark proposed in [59].

## 6.2 Image dataset with known MI

**Setups**. We next consider the benchmark [59], which contains correlated images $X$ and $Y$; see Figure 4. Here $X \in \mathbb{R}^{16 \times 16}$ and $Y \in \mathbb{R}^{16 \times 16}$, and the ground truth $I(X;Y)$ is known for this dataset. Following recent works [59, 3], we preprocess these high-dimensional image data by an autoencoder $e : \mathbb{R}^{16 \times 16} \to \mathbb{R}^{d'}$, which proves effective in reducing data dimensionality while preserving key information. The quality of such compression w.r.t $d'$ is analyzed theoretically and empirically in Appendix A5 and B2, based on which we set $d' = 16$. A total number of $10,000$ data is used. Note that while the dependence between $X$ and $Y$ are Gaussian for this dataset [59], the dependence structure for the compressed data can be non-Gaussian even if the compression is near-lossless.

**Results**. Table 1 compares the performance of different MI estimators on this task. Our estimator consistently outperforms the recent $\mathcal{N}$-MIENF estimator on this dataset, and it shows highly competitive performance against discriminative methods. However, our method performs slightly worse than discriminative methods in the Rectangles case. One reason why our approach loses to discriminative approaches in the Rectangles case may be that the underlying dependence structure of the preprocessed data is highly complex in this case, which is difficult to model effectively with a single vector Gaussian copula or even a reasonable mixture of such copulas. Discriminative methods, on the contrary, adopt a neural network-based parameterization of the vector copula, being inherently more flexible. These results highlight the limitation of model-based parameterization of the vector copula density in certain cases. Nonetheless, our estimator still provides a highly reliable estimate.

## 6.3 Embeddings of language models

**Setups**. We further consider a real-world dataset in natural language processing. It consists of pairs of embeddings from a language model (LM) [60, 61] computed on the IMDB dataset [62], which contains negative or positive movie comments; see Table 2. The ground truth MI of this dataset is unknown, but it can be computed numerically accurately; see Appendix B1. A total number of $n = 4 \times 10^3$ data are used. Similar to the previous task, we preprocess data by an autoencoder $e : \mathbb{R}^{d_{\text{LM}}} \to \mathbb{R}^{16}$, with $d_{\text{LM}}$ being the dimensionality of the LM's embeddings. The quality of such compression is empirically studied in Appendix B2, which is near-lossless.

**Results**. Table 3 summarizes the results for this dataset. In this scenario, where the underlying mutual information (MI) is relatively low, our method does not show a significant advantage over discriminative methods. This is likely because for this dataset, the high-discrepancy issue [40, 41, 29, 32] is not significant, and discriminative methods offer a more flexible parameterization of the vector copula density $c$ than our method (see Section 5). Nonetheless, our method still provides an estimate close to discriminative methods, and it significantly outperforms the generative method $\mathcal{N}$-MIENF.

## 6.4 Further analysis and ablation studies

We conduct further analysis on the effect of *model selection* and *separate learning* in Appendix B2.

|   | $X$ | $Y$ |
|---|---|---|
| 1 | (positive) I thought this was a wonderful way to spend time on ... | (positive) If you like original gut wrenching laughter you will like ... |
| 2 | (negative) So im not a big fan of Boll's work but then ... | (positive) This a fantastic movie of three prisoners who become famous... |

Table 2: The text benchmark, which contains reviews of positive or negative movie comments.

| Method | $I(X;Y) \approx 2.1$ | $I(X;Y) \approx 0.9$ |
|---|---|---|
| MINE | $1.83 \pm 0.04$ | $0.71 \pm 0.05$ |
| InfoNCE | $1.64 \pm 0.09$ | $0.70 \pm 0.06$ |
| MRE | $1.72 \pm 0.07$ | $1.23 \pm 0.02$ |
| $\mathcal{N}$-MIENF | $0.91 \pm 0.05$ | $0.43 \pm 0.03$ |
| VCE | $\mathbf{2.01 \pm 0.04}$ | $\mathbf{0.83 \pm 0.01}$ |

**(a) Llama-3 13B**

| Method | $I(X;Y) \approx 1.5$ | $I(X;Y) \approx 0.2$ |
|---|---|---|
| MINE | $\mathbf{1.42 \pm 0.04}$ | $0.18 \pm 0.02$ |
| InfoNCE | $1.41 \pm 0.03$ | $0.19 \pm 0.04$ |
| MRE | $1.23 \pm 0.09$ | $0.31 \pm 0.09$ |
| $\mathcal{N}$-MIENF | $0.73 \pm 0.03$ | $0.11 \pm 0.02$ |
| VCE | $1.22 \pm 0.02$ | $\mathbf{0.19 \pm 0.02}$ |

**(b) BERT**

Table 3: Comparing different MI estimators on the text dataset. Left: evaluation on the embeddings of Llama-3 13B model [61]. Right: evaluation on the embeddings of a BERT model [60].

# 7 Conclusion

In this work, we introduced a new mutual information (MI) estimator grounded in recent vector copula theory. A fundamental difference to existing approaches is the explicit disentanglement of marginal distributions and dependence structure in our method. This separation enables more flexible and fine-grained modeling, avoiding the pitfalls of both overly simplistic or excessively complex approaches, and reducing overall learning difficulty via strategic factorization of the original estimation problem. Extensive experiments demonstrate our method's effectiveness and robustness.

Beyond the development of practical estimator, our research also offers fresh perspectives on MI estimation. By viewing PMI as a density rather than a density ratio, we open new avenues for modeling. Additionally, our approach to vector rank computation generalizes the classical copula transformation and holds promise as a versatile preprocessing step for a broad range of MI estimators. Finally, by reinterpreting existing estimators through the lens of vector copula theory, we obtain new insights into the parameterization and learning of different estimators and the underlying trade-offs.

Copulas have been widely used for MI estimate [63, 55, 64, 56, 65, 33, 51, 45, 66]. Existing methods primarily focus on *classic copulas*, where the copula transformation is applied independently to each univariate marginal to better account for the marginal-invariant property of MI. This strategy has been shown to improve accuracy and reduce variance [56, 65]. We go one step further by using *vector copulas*, where the transformation jointly considers all dimensions of the multivariate marginals. This can be seen as a generalization of classic copula transformation, where we not only consider MI's invariance to *element-wise* bijections but also to *any* diffeomorphisms. Another key difference lies in that these works still treat PMI as a density ratio, whereas our work treats PMI as a density.

We note that, while powerful, our estimator is not a panacea. One limitation of our method is that it relies on the two marginal distributions to be reasonably modeled. While marginal distributions are far easier to learn than the joint distribution, they can still be challenging to learn for high-dimensional data e.g., images. Fortunately, dimensionality reduction techniques [3, 13] help to mitigate this issue. Another limitation lies in the flexibility of our model-based parameterization of vector copula, which can be less flexible than neural network methods. However, as our method strikes a good trade-off between complexity and capacity across diverse cases, we consider it as a highly competitive method.

## Acknowledgments

AW acknowledges supports from a Turing AI Fellowship under grant EP/V025279/1, the Alan Turing Institute, and the Leverhulme Trust via the Leverhulme Centre for the Future of Intelligence. MUG was supported in part by Gen AI - the AI Hub for Generative Models, funded by EPSRC (EP/Y028805/1). YC acknowledge supports from the Cambridge Trust and the Qualcomm Innovation Fellowship. ZO acknowledges supports from Lee Family Scholarship via Imperial College London.

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
