# OpenReview forum: "Neural Mutual Information Estimation with Vector Copulas"
_NeurIPS.cc/2025/Conference — NeurIPS 2025 poster_

### Official Review · Reviewer_44kJ · 2025-06-21

**Clarity:** 3
**Significance:** 3
**Originality:** 2
**Rating:** 4
**Confidence:** 5

**Summary:**

The paper utilizes the concept of vector copula introduced in [27] to develop a generative MI estimator (VCE). The authors discussed important properties of VCE such as consistency and convergence. They also commented on desirable features and interesting concepts such as separate learning/modeling and complexity-capacity trade-off. Moreover, given the interpretation of the PMI as a density rather than a density ratio, the authors studied neural-network free methodologies (e.g., MLE) to model the vector copula. The VCE estimator is compared with several established estimators (both discriminative and generative). Experiments show promising results for low-dimensional data and for low values of MI.

**Questions:**

Overall, I found this paper interesting and well-motivated, and I would like the authors to address the following questions to help clarify the contributions and potential impact:

1. **Comparison with SOTA Methods.** Authors selected MRE [35] and MINDE [16] as SOTA estimators for discriminative and generative approaches, respectively. This is definitely arguable, at least for the discriminative approach as also SMILE [29] and $f$-DIME [36] are robust estimators. Moreover, [29] and [50] introduced common benchmarks and evaluation techniques which in this paper are not fully considered. Could the authors include richer comparisons with these methods? This would provide a more comprehensive evaluation and help establish where VCE stands relative to current best practices. Additionally, could the authors provide clear recommendations on when VCE should be preferred over other methods?

2. **Bias-Variance Analysis.** The paper lacks discussion about bias-variance trade-offs, which is crucial for MI estimators. Could the authors provide a theoretical analysis or experimental evaluation of how the VCE estimator's bias and variance scale with sample size? This would help clarify the estimator's statistical properties and practical applicability.

3. **Neural Network Parameterization.** The paper focuses on neural-network free methodologies for modeling the vector copula density. Given that the error depends on the KL divergence between $c$ and $c'$, could the authors provide at least one concrete example comparing neural network parameterization vs. classical methods for the vector copula density estimation? This would help understand the trade-offs and potential benefits of different modeling approaches, and could reveal whether the neural-network free approach is indeed still a valid path even for high dimensionality.

4. **High MI and High Dimensionality Scenarios.** The experiments focus on low-dimensional data and low MI values, but the most challenging problems in MI estimation occur with high-dimensional data and high MI values. Could the authors extend their experiments to include high-dimensional scenarios (e.g., $d > 50$) and high MI values? This would demonstrate the method can be used for real-world applications and could significantly increase the significance of the paper if the VCE estimator shows advantages in these challenging regimes.

5. **Comparison with Related Copula-based Methods.** The papers [R1, R2] briefly discuss the advantages of modeling the density ratio (not seen as a density) via classic copulas for the MI estimation task. I think it might be worthy add a comparison (at least written) with these papers. Similarly, can the authors elaborate more (perhaps add some sentences if not in the main also in the appendix about the difference between this paper and the paper in [40] as they both study the MI as negative differential entropy, but the former with the concept of vector copula?) I think it might help the reader better understand the novelty of this paper.

6. **Double-Step Approach Validation.** The paper proposes a two-step approach: flow matching to obtain vector ranks followed by classical parametric methods (the authors mentioned about the possibility to train neural networks as well) to estimate the vector copula density. However, marginal distribution learning via flow matching generative models could not reach convergence and classical parametric methods are known to suffer from various limitations (e.g., model misspecification, curse of dimensionality). Could the authors provide a more thorough analysis of how this double-step approach actually improves upon existing single-step methods? Specifically, could they demonstrate quantitatively that the benefits of the conceptual separation outweigh the potential errors introduced by the two-step process? This would help validate whether the proposed approach genuinely advances the state-of-the-art or simply trades one set of problems for another.

[R1] Samo, Yves-Laurent Kom. "MIND: Inductive Mutual Information Estimation, A Convex Maximum-Entropy Copula Approach." arXiv preprint arXiv:2102.13182 (2021).

[R2] Letizia, Nunzio A., and Andrea M. Tonello. "Copula density neural estimation." arXiv preprint arXiv:2211.15353 (2022).

**Ethical Concerns:**

["NO or VERY MINOR ethics concerns only"]

**Final Justification:**

The Authors have adequately addressed all my concerns, which are now resolved.

**Limitations:**

Yes

**Quality:**

3

**Strengths And Weaknesses:**

The paper is technically sound and the theoretical analysis includes consistency and convergence/error proofs for the VCE estimator. The authors provide various experimental setups demonstrating the applicability of the method across different scenarios. The work is well-organized with clear sections and an accurate literature review, including a clear explanation of the vector copula concept and methodology.

The main contribution of the paper is the formalization of a novel MI estimator based on the concept of vector copula, with the validation of desirable properties. Another interesting contribution lies in interpreting the PMI as density itself rather than density ratio (which follows [27]), which could advance the field and shed light on how to treat the PMI. The authors are honest about limitations, which are clearly stated.

---

> ### Author Rebuttal · Authors · 2025-07-25
>
> Thank you for your very insightful comments and the constructive criticisms. We believe many criticisms (for instance, no high-MI/high-dimensionality experiments, improper evaluation) were due to some misunderstandings, which we clarify below. Many of the requested comparisons/analyses can also be found in the provided appendix. Meanwhile, we have included new results and theories, as well as properly discussed related works to fully address your concerns.
>
> &nbsp;
>
> &nbsp;
>
>
>
> ### **1a. Comparison with STOA Discriminative Methods**
>
> We address your concerns on baselines from two aspects: (a) clarifying our baseline choice and (b) futher comparing with the suggested baselines.
>
> - (**MRE is a competitive baseline**) We fully agree that $f$-DIME and SMILE are STOA estimators, as we will note in our paper. That said, MRE is also highly competitive: it is verified on high-dimensional non-Gaussian data (80D) and high MI regimes (20+nats). Importantly, MRE shares key similarities with GAN-DIME---both methods estimate the ratio via classifier training and avoid computing the partition function. In practice, we found them almost equally strong, so we mainly compare with MRE for space efficiency.
> - (**Further comparison to SMILE**)  Following your suggestion, we have also comapred to SMILE; see the table below. The results highlight the advantanges of our method. A more complete comparison will be included in the manuscript.
>
> Table 1. Comparing SMILE vs VCE with 64D data. Results are collected from 8 independent runs. Code available in the provided code repo.
>
> |         &nbsp;&nbsp; |   &nbsp; Spiral &nbsp;   |   &nbsp; Cubic &nbsp;   |  &nbsp; Student-t &nbsp;  |   &nbsp; MoG  &nbsp;   |    &nbsp; MoT &nbsp;    |
> |-------|:----------:|:----------:|:-----------:|:----------:|:----------:|
> | SMILE-1 &nbsp;&nbsp; | &nbsp;1.54 ± 0.05&nbsp;| &nbsp;0.01 ± 0.02&nbsp;| &nbsp;1.82 ± 0.05&nbsp;| &nbsp;1.90 ± 0.06&nbsp;| &nbsp;1.14 ± 0.03&nbsp;|
> | SMILE-10 &nbsp;&nbsp; | &nbsp;3.22 ± 1.15&nbsp;| &nbsp;1.07 ± 0.08&nbsp;| &nbsp;5.12 ± 0.17&nbsp;| &nbsp;6.81 ± 0.20&nbsp;| &nbsp;3.22 ± 0.12&nbsp;|
> | SMILE-25 &nbsp;&nbsp; | &nbsp;2.72 ± 1.45&nbsp;| &nbsp;6.21 ± 0.34&nbsp;| &nbsp;4.60 ± 0.47&nbsp;| &nbsp;6.75 ± 0.20&nbsp;| &nbsp;3.08 ± 0.09&nbsp;|
> |   VCE   &nbsp;&nbsp; | &nbsp;8.27 ± 0.25&nbsp;| &nbsp;10.00 ± 0.30&nbsp;| &nbsp;9.76 ± 0.37&nbsp;| &nbsp;12.49 ± 0.37&nbsp;| &nbsp;7.20 ± 0.22&nbsp;|
> |    GT   &nbsp;&nbsp; | &nbsp;10.77&nbsp;      | &nbsp;10.77&nbsp;      | &nbsp;12.46&nbsp;      | &nbsp;13.19&nbsp;      | &nbsp;6.90&nbsp;      |
> |  |  |  |  |  |  |
>
> If necessary, we are fully happy to replace MRE with $f$-DIME or SMILE in our main text. We wish it understandable that it was unrealistic to include every baseline due to space limit.
>
> &nbsp;
>
> &nbsp;
>
> ### **1b. Issues with Benchmark & Evaluation Protocol** ###
>
> We clarify that **our benchmark and protocols highly overlap with SOTA** benchmarks [44, 50, 51] and recent estimators (and **may even be richer**). See Table 2 for a comparsion.
>
> Table 2. Comparing the benchmarks and evaluation protocols in different works.
>
> |  | &nbsp;  **Synthetic distributions** | **Real-world dataset**  | **Evaluation protocol**|
> | --- | --- | --- | --- |
> | **$f$-DIME** | &nbsp;  Gaussian, cubic, half cube, asinh, uniform, student-t, swiss roll | MNIST & FMNIST | Direct comparison with GT + SC test$^†$
> | **$\mathcal{N}$-MIENF** | &nbsp;  Gaussian, cubic, asinh+student-t, uniform, student-t  &nbsp; | Images with known MI | Direct comparison with GT |
> | **VCE** | &nbsp; Gaussian, cubic, tanh-and-exp, student-t, swiss roll, spiral, mixture distributions &nbsp; | Images with known MI, LLM embeddings &nbsp; | Direct comparison with GT |
> |  |  |  | |
>
> $^†$We do not employ the self-consistency (SC) test from [29], as all our test cases (including images and texts) have known and controllable ground truth (GT), requring not using this test.
>
>
>
>
>
> &nbsp;
>
> &nbsp;
>
>
> ### **2. Bias-and-variance Analysis** ###
>
> - (**Empirical analysis**) Following your valuable suggestion, we have now included an empirical variance analysis of VCE. See Table 3 for an example.
>
> Table 3. Empirical variance of VCE with 64D data w.r.t sample size $n$. Reported values are the std across 16 randomly sampled batches.
>
> |                | n=50        | n=100       | n=500       | n=1000      | n=2000      | n=5000      | GT         |
> | -------------- | ----------- | ----------- | ----------- | ----------- | ----------- | ----------- | ---------- |
> | Student-t (v=1) &nbsp; &nbsp;| 0.6242      | 0.4764      | 0.1814      | 0.1787      | 0.0507      | 0.0342      | 12.0022    |
> | Spiral       &nbsp; &nbsp;  | 1.4173      | 0.3176      | 0.3114      | 0.2741      | 0.1451      | 0.1218      | 10.7725    |
> |||||||||
>
>
> - (**Theoretical analysis**). We also derive a new proposition analyzing how the number of mixture components $k$ governs the bias-variance trade-off in vector copula modeling:
>
> (*Proposition 5*) For large sample size $n$, the expected KL divergence between the true vector copula $c$ and its $k$-component vector Gaussian copula mixture estimate $\hat{c}_k$ satisfies
>
> $
> \mathbb{E} \left[ \mathrm{KL}[c || \hat{c}_k] \right] \leq C_1 \cdot k^{-\frac{2\alpha}{d}} + C_2 \cdot \frac{k d \log (C_3 n)}{n},
> $
>
> where $\alpha, C_1, C_2, C_3 > 0$ are constants depending on the true vector copula solely, $d$ is the data dimension and $n$ is the sample size. Here, the 1st term on the RHS quantifies the **bias**, which decreases as $k$ grows and is irrelevant to the sample size $n$. The 2nd term on the RHS corresponds to the **variance**, which increases as $k$ grows and decreases as $n$ grows.
>
>
>
> &nbsp;
>
> &nbsp;
>
> ### **3. Neural Parameterization vs Model-based Parameterization**
>
> Thank you for highlighting this comparison --- we are on the same page! **Figure 1 in Appendix B2** precisesly perform this comparison. In the figure, we compare **VCE (model-based parameterization)** and **VCE’ (neural parameterization)** in diverse settings. The results clearly show the advantange of model-based parameterization in the considered settings.
>
> We fully understand that the reviewer is skeptical regarding the applicability of classic methods in high-dimensional space. However, please note that we are not applying classic methods directly in the original space, but rather a transformed space learned by flow models. This transformation bridges the gap, making classic method viable in this space.
>
> &nbsp;
>
> &nbsp;
>
>
> ### **4. High MI and High Dimensionality (d>50) Scenarios**
>
> We clarify that our original submission actually include both these cases (though they might not be so easy to notice due to our presentation); see Table 4 below for a summary. Our method demonstrates either significant advantange or highly competitive performance in both cases.
>
> Table 4. MI and data dimesnionalities considered in this work.
>
> |  | Detail &nbsp; | Evidence |
> | --- | --- | --- |
> | High MI &nbsp; | 14nats non-Gaussian data, 7nats image and text data &nbsp; | Figure 1a-1e cases $\rho=0.8$, Figure 2a-2e cases $d=64$  |
> | High dimensionality &nbsp; | 64D imcompressible data, 32x32 images*, 512D text embeddings*  &nbsp; | Figure 2a-2e cases $d=64$, Section 6.2, Section 6.3 |
> ||||
>
> *For compressible data, we leverage injective flow (implemented as autoencoder + latent flow). The same autoencoder is used for all other estimators for fairness.
>
>
> &nbsp;
>
> &nbsp;
>
> ### **5. Comparison to Related Copula Methods** ###
>
> - **Additional paragraph comparing our work with [R1, R2]**
>
> “*Copula has been widely used for MI estimate [R1, R2, other works]. Existing methods primarily focus on classic copulas, where the copula transformation is applied independently to each univariate marginal. This strategy has been shown to improve accuracy and reduce variance [R2]. We go one step further by using vector copulas, where the transformation jointly considers all dimensions of the multivariate marginals. This can be seen as a generalization of classic copula transformation, where we not only consider MI's invariance to element-wise bijections but also to any bijective function. Another key difference lies in that existing works [R1, R2, …] treats PMI as a density ratio, whereas our work treats PMI as a density.*“
>
> - **Texts clarifying the difference between our work and [40]**
>
> “*Similar to our work, prior work [40] also interprets MI as the differential entropy of copula. However, building upon classic copulas, their result only holds for scalar marginals, and generalizing their result to vector marginals require non-trivial formulation and derivation. Our work in contrast holds for vector marginals with any dimensionalities by leveraging vector copulas.*”
>
> We promise that all these discussions will be included in the final version. Thank you once gain for highlighting this important comparison.
>
> &nbsp;
>
> &nbsp;
>
>
>
>
>
> ### **6. Double Step Approach Validation** ###
>
> We are on the same page again --- the ablation study **Joint Learning vs Separate Learning in Appendix B2** precisely performs this quantitative analysis. The results clearly demonstrate the benefits of the double step approach over single step.
>
> Note that even if the learned vector ranks are imperfect, double step learning can still be useful. This is because the first step will not alter the underlying MI. In this regard, computing vector ranks can be viewed as an informed data-preprocessing step to improve MI estimation --- akin to the scalar rank transformation used in the CODINE paper [R2].
>
> &nbsp;
>
> &nbsp;
>
>
> **Summary**: VCE is a robust and accurate MI estimator across diverse settings, including high-MI regimes (e.g. 14nats), high-dimensional data (e.g., 64D incompressible data, 512D text embeddings), long tails, non-Gaussian dependencies, heteogeneous marginals, etc. However, it may be is less suitable for incompressible data with very high dimensionality (>128D).

---

> ### Author Response · Authors · 2025-08-03
>
> Dear reviewer 44kJ,
>
> Thank you once again for your thoughtful review and for signing the Mandatory Acknowledgement. Your inputs have been highly valuable. We would be grateful to learn from you in details whether our rebuttal has addressed your concerns, particularly those stemming from **potential misunderstandings** (e.g. no high MI/high-dim experiments, benchmark choices) and **insufficient prior work discussion**. This would greatly help us improve our work regardless of the final decision.
>
> Best regards, Authors

---

> ### Comment · Reviewer_44kJ · 2025-08-04
> **Thanks for the discussion**
>
> Dear Authors,
>
> Thanks a lot for having clarified some of the initial doubts I had while reading the paper. As a matter of fact, some of these were indeed already solved in the paper and in the appendix. I'm confident the new version will leave no room for doubts.
>
> As all my concerns are now resolved, I'm willing to raise my score to 4. Looking forward to reading the revised version of this paper.

---

> > ### Author Response · Authors · 2025-08-04
> > **Thank you!**
> >
> > Thank you very much for your confirmation! The discussion has been truly fruitful. We will make sure all the modifications are incorporated in the new version. We sincerely appreciate your help in improving our work.

---

### Official Review · Reviewer_Gion · 2025-06-27

**Clarity:** 3
**Significance:** 3
**Originality:** 2
**Rating:** 4
**Confidence:** 1

**Summary:**

The paper  proposes an copula based approach (and algorithm) to compute  mutual information.
Experimental results show that the proposed approach can outperform alternative algorithms.

**Questions:**

none

**Ethical Concerns:**

["NO or VERY MINOR ethics concerns only"]

**Final Justification:**

NA

**Limitations:**

yes

**Quality:**

3

**Strengths And Weaknesses:**

**strengths:**

- Extensive comparison with existing algorithms.
- demonstrate good accuracy
- code provided


**weaknesses:**
- Figures too small (Fig.1/2)
- Appendices missing (?)
- Table 2 not needed (?)

---

> ### Author Rebuttal · Authors · 2025-07-25
>
> Thank you for your time and your work in reviewing our submission! Although being concise, your feedback is still valuable in improving our work. We truly appreciate your recognition of our work's merits, including our extensive evaluation, the reproducibility, and the overall strong performance. Below, we address your concerns.
>
> - **Figures too small**: We sincerely apologize for this. In the revised version, we have (a) resized all figures and (b) increased the font size to improve clarity.
> - **Appendix missing**: The appendix is actually provided as a separate file via the ‘Supplementary Materials’ link above. We have now merged it with the main text for better clarity.
> - **Table 2 may be redundant**: This could be true, but Table 2 may help readers to get a sense of the text dataset, as it is a relatively new dataset to the MI community.
>
> We hope that we have addressed your concerns. Thank you again for reviewing our work. Please don’t hesitate to let us know if there are any additional comments or suggestions!

---

### Official Review · Reviewer_A2bS · 2025-06-29

**Clarity:** 3
**Significance:** 4
**Originality:** 3
**Rating:** 5
**Confidence:** 4

**Summary:**

The paper shows that MI can be recast as a vector copula estimation task, thereby generalising the previous results where the link between copulas and MI was established for univariate variables. A vector copula method is proposed to estimate MI in two steps, first by fitting separate flow models on each set of variables X and Y, and secondly, learning the dependence between X and Y with a mixture of Gaussian vector copulas. Theoretical arguments motivate the approach, and experiments show its competitiveness compared to existing methods for MI estimation.

**Questions:**

Overall, the paper is persuasive about the usefulness of vector copulas for MI. However, I am unsure about the practical implementation proposed in this paper. Before I can confidently advocate the acceptance of this paper, **I would like to see the first four weaknesses addressed**. Furthermore, critical questions I would kindly ask to be answered include:

- Could you provide, at least in the appendix, some evidence of the obtained vector ranks being $\mathcal{N}(\mathbf{0},\mathbf{I})$ for your experiments? Maybe a test for Gaussianity/independence, or showing the univariate ranks of the representations, or the covariance matrices, or metrics derived from these?

- Are there other ways for obtaining vector ranks that you have considered? For instance, in the 2023 Vector Copulas paper, the authors consider an optimal transport problem solved through linear programming. How would these compare, computationally and results-wise or are they not appropriate to be used here?

---

Below are minor points and typos that I believe should be corrected before this paper can be ready for acceptance, as well as a few questions I am curious about but that I do not need to be answered if time is too short:

**Typos and notation**

- L24 Do you mean 'suffer from requiring an exponentially large...'?
- Definition 1, there is a missing dot to the statement?
- Notation in L98-99, I find it more common to use $\Phi$ for the CDF and reserve $\phi$ for the PDF. A possible alternative to notation could be to use $\Phi_d$ to distinguish the element-wise CDF from the joint?
- L101 $\epsilon_{d_X} $ should be $\epsilon_{\leq d_X} $ ?
- L149 $c$ *as* a mixture?
- L152 stricture -> structure
- L158, of *a* copula?
- Appendix L55, should say by Lemma 2 instead?
- Appendix L89, these combined *results* lead to?
- Appendix, define $\Sigma$ in Lemma 3.
- Throughout your proofs in the Appendix, at APDX L133-134, and more generally, to be more respectful to readers, I would replace 'it is clear' by the appropriate reason. Eg ' we identify this as a Gaussian density', 'By definition, it follows that...'.
- Figure 1 and 2 could use bigger fonts for the axis and titles, same with the appendix.
- L266 known *to be* challenging?
- L297, please write what $d_L$ is equal to in your experiment.

**Curiosity**

- L194, Does a second-order Taylor expansion imply second-order dependencies? For instance, in classical copulas, Gaussians satisfy the simplifying assumption of vines - would this be the same for the vector Gaussian copula? Then, does this mean your V-Gaussian mixture is simplified?
- L235, Do you test for or show in one of your examples? Time permitting, it would perhaps be nice to see.
- L286-287, Could this also be because the neural estimators do not need to be normalised and might overestimate the true MI?

**Ethical Concerns:**

["NO or VERY MINOR ethics concerns only"]

**Final Justification:**

The Authors clearly addressed all my concerns. They proposed additional diagnostics and a discussion of some related work, and promised to include those in the final version of the paper. I was already positive about this paper with a rating of 4, and I am increasing my rating to 5 after the discussion, as I confidently believe this paper is of sufficient quality and merit to be published at NeurIPS.

**Limitations:**

Limitations are generally addressed. However, I would appreciate more emphasis on the fact that vector ranks need to be standard Gaussians in the proposed approach, as it is critical for the theory and method to work. I believe things that would ameliorate this include putting more emphasis on this condition being met when relevant, and providing or suggesting some diagnostics or tests to check for this assumption being respected.

**Quality:**

3

**Strengths And Weaknesses:**

I believe the paper offers a promising way forward in MI estimation, differentiating itself from the more common density ratio approaches that could lead to unnormalized estimates. The separation of learning vector marginals first and the vector copula second allows great control over the method’s complexity. Theoretical results are generally good (except Prop 4 which I have doubts about) and provide good arguments for the adoption of this method. Experiments are also convincing.

---

**Strengths**
- The paper’s main novelty in Theorem 2 offers a theoretically grounded way of estimating MI, which possibly provides a sound direction for future progress in this field.
- Their proposed estimator offers some good flexibility in terms of selecting an appropriate model capacity for the perceived complexity of a task, as well as targeting whether to add complexity to the vector marginals and/or the vector copula.
- I found Proposition 3 to be very good motivation for the proposed modelling approach, exemplifying the capacity-complexity trade-off.

---

While the first part of the paper, up to L126 had me convinced by providing a sound motivation, I am very uncertain about Section 3.1, which holds me back from fully believing in the proposed approach. I also have some comments on some aspects of training feasibility, relations to previous work, the vector copula mixture, and the meaning of Proposition 4.

**Weaknesses**

- I am particularly sceptical about the obtained vector ranks as this is crucial for your theory to hold, and for your estimates to be meaningful:

To use a vector copula, you need vector ranks. To obtain those, you choose flow models. While it certainly is easier to estimate two separate flows in $d_X$ and $d_Y$ dimensions compared to one big flow in $d_X+D_Y$, this is still not at all obvious. L133-134 is crucial to your method, yet it is not emphasised enough and checked for in my opinion.
- I am not sure joint learning of vector copulas is a well-posed problem:

At the bottom of page 3, and in the Appendix L251, you suggest learning vector ranks with the vector copula jointly. But is this well defined? I believe [TACTiS-2: Better, Faster, Simpler Attentional Copulas for Multivariate Time Series, ICLR 2024] showed that joint optimisation is ill-defined for regular copulas – would the same not be true here? Your experiments in the Appendix seem to suggest so…

- It is not clear to me how the vector copula mixture obtains meaningfully different mixture components $c_k$:

As you optimise the MLE on observed vector ranks, how do you ensure that each mixture component will not obtain the same representation? There are no real details about the implementation you use that I could spot.

- Two relevant previous works should be discussed in relevant sections of the text:

In Algorithm 2, your approach of using a DRE to estimate the ratio between a simple vector copula and the true vector copula has previously been used in the context of copulas in [Your copula is a classifier in disguise: classification-based copula density estimation, AISTATS 2025], in their Section 3, and I believe this should be acknowledged. Furthermore, in your Section 5, this same paper discusses many similar points to L207-220 in their Section 6. A second reference that I would like to see discussed is [Towards a universal representation of statistical dependence, by Gery  Geenens 2023], where the relationship between copulas and MI is discussed, notably in their Section 6 showing MI is a measure on non-independence while copulas fully capture dependence.

- Proposition 4 I believe is a little misleading:

In your proof in the Appendix, before concluding in L150, you end up with a representation of an independent Gaussian in $d_Y+d_X$ dimensions. While this is a Gaussian, it is fully independent. How is your statement different from saying that an independent copula is a Gaussian copula because you can set the correlation matrix $\Sigma$ to be diagonal while interpreting your data generating process to be with Gaussian variables? All this is saying is that a Gaussian vector copula can also capture the independence vector copula, at which point I fail to see the relevance of this statement, or am I missing something?

---


**Quality**: The motivation for the method is well articulated and supported with appropriate theoretical results, and experiments are rather exhausting covering multiple setups. However I am not sure about practical details of the proposed method, which I believe have to be addressed more.

**Clarity**: The submission is overall clearly written and well organised with the only details missing being about the copula mixture and missing discussions of related work.

**Significance**: I believe this paper presents good arguments for a possible shift in focus within the MI community, offering a new objective, providing links within the literature, and reinterpreting existing models for new insights. It also provides motivation to develop new vector copula models with MI constituting a useful application for them.

**Originality**: New insights and properties of existing work are discussed. But some related work that I believe should be discussed is omitted. The introduction of vector copulas for MI, while not fully novel from the copula perspective, is motivated well and backed up by theoretical guarantees.

---

> ### Author Rebuttal · Authors · 2025-07-26
>
> Thank you for your high-quality review and the detailed feedbacks! We are deeply grateful for your recognition of our work’s theoretical maturity, potential impact in creating a focus shift in MI community and our comprehensive evaluation. Below, we address your concerns, particular those regarding **the quality of the obtained vector ranks** and **related works**.
>
> &nbsp;
>
> &nbsp;
>
> ### **W1. Sceptical about the quality of the obtained vector ranks** ###
>
> This is a valid concern. We address it from two aspects:
>
> - (**Guaranteeing uniformity**) Each univariate rank  $\mathbf{\hat{u}}_d$ is guaranteed to be *perfectly* uniform in our method. Specifically, when computing the vector ranks, we applied element-wised empirical rank rather than Gaussian CDF; see the provided code. The description in the manuscript was indeed inaccurate, and we sincerely apologise for this inaccuracy.
> - (**Testing independence**) Following the reviewer’s insightful suggestion, we have performed a diagnostic to validate whether dimensions  $\mathbf{\hat{u}}_i$,  $\mathbf{\hat{u}}_j$ are independent. Tables 1 and 2 summarize the results. It is clear that $\mathbf{\hat{u}}_i$,  $\mathbf{\hat{u}}_j$ are indeed near-independent in moderate-to-high dimensional settings. We will include this diagnostic in the revised manuscript.
>
> Table 1. (Student-d distribution case). Inspecting the non-diagonal elements $\Sigma_{ij}$ in the covariance matrix $\Sigma$ of the estimated vector rank $ \mathbf{\hat{u}}$.
>
> |  | &nbsp; mean of $\Sigma_{ij}$ &nbsp; | &nbsp; 5% quantile of $\Sigma_{ij}$ &nbsp; | &nbsp; 95% quantile of $\Sigma_{ij}$ &nbsp; |
> | --- | --- | --- | --- |
> | d=32 | &nbsp; 0.007 | &nbsp; -0.042 | &nbsp; 0.031 |
> | d=64 | &nbsp; 0.012 | &nbsp; -0.048 | &nbsp; 0.042 |
> |  |  |  |  |
>
> Table 2. (Spiral transformation case). Inspecting the non-diagonal elements $\Sigma_{ij}$ in the covariance matrix $\Sigma$ of the estimated vector rank $ \mathbf{\hat{u}}$.
>
> |  | &nbsp; mean of $\Sigma_{ij}$ &nbsp; | &nbsp; 5% quantile of $\Sigma_{ij}$ &nbsp; | &nbsp; 95% quantile of $\Sigma_{ij}$ &nbsp;|
> | --- | --- | --- | --- |
> | d=32 | &nbsp; 0.007 | &nbsp; -0.053 | &nbsp; 0.055 |
> | d=64 | &nbsp; 0.009 | &nbsp; -0.079 | &nbsp; 0.066 |
> |  |  |  |  |
>
> While the above results indicate **good vector rank quality (perfect element-wise uniformity + low correlation)**, quality degration may still occur. The above test serves as a highly useful diagnostic for quality assessment. We sincerely thank the reviewer for prompting this checking, which has been instrumental in improving the rigor and practicality of our work.
>
> &nbsp;
>
> &nbsp;
>
> ### **W2. Joint learning of vector copulas is not a well-posed problem** ###
>
> We are on the same page! **Our prior mention of joint learning was merely for completeness, not to recommend it**. In fact, separate learning is a key design in our method, as motivated by l120–l126 and justified by the ablation study “Joint Learning vs. Separate Learning” in Appendix B. The reviewer's reference further supports this design.
>
>
>
> &nbsp;
>
> &nbsp;
>
> ### **W3. How to obtain meaningfully different mixture components $c_k$ in the vector copula** ###
>
> The answer is simple: **just randomly initialize the parameters of each mixture component**, following standard practice in Gaussian mixture modeling. This is analogous to how neural networks are initialized to avoid homogeneous weight configurations. If the true dependence is non-Gaussian, MLE should learn diverse components to attain a high likelihood.
>
> &nbsp;
>
> &nbsp;
>
> ### **W4. Relevant previous works missing** ###
>
> Thank you very much for pointing us to these highly relevant works, which we have now discussed properly in the paper:
>
> - (**Acknowledging [A]**). When introducing VCE’ (reference-based parameterization of vector copula), we cite [A] to explicitly acknowledge that similar method has been explored for classic copula. We have also related our Section 5 to the analysis in [B], thereby drawing a connection between the two works.
> - (**Discussing [B]**). We have now included a new paragraph discussing existing copula-based MI estimators, where both [A][B] and other relevant works are included.
>
> [A] Your copula is a classifier in disguise: classification-based copula density estimation, AISTATS 2025
>
> [B] Towards a universal representation of statistical dependence, 2302.08151.
>
> &nbsp;
>
> &nbsp;
>
>
> ### **W5. Confusion on Proposition 4** ###
> We feel that we are on the same page --- Proposition 4 indeed suggests that *a vector Gaussian copula model can perfectly represent any independent distribution* $p'(\mathbf{x}, \mathbf{y}) = p(\mathbf{x})p(\mathbf{y})$.
>
>
> &nbsp;
>
> &nbsp;
>
>
> ### **Questions** ###
> - (**The quality of the obtained vector ranks?**) Thank you for this great suggestion. We have now included a diagnostic for assessing the quality of the learned vector rank $\hat{\mathbf{u}}$; please see our response to W1 above. More advanced tests can also be used, but we consider it as future works.
>
> - (**Other ways for obtaining vector ranks?**) The linear programming approach from the original vector copula paper can also be used, but we found their method computationally expensive for even moderate dimensional variables (e.g. 30D), so we use the flow-based approach instead.
>
>
> &nbsp;
>
> &nbsp;
>
>
> ### **Curiosity**
> > **Is vector Gaussian copula mixture a simplification of vines?**
>
> This is an inspiring ask! It could be, but a careful check is needed, as we work with high-dimensional vector Gaussian copula rather than the commonly used bivariate copula in vines.
>
> > **Do you test for or show in one of your examples that $\mathcal{N}$-MIENF / DINE-Gaussian indeed learns a vector Gaussian copula?**
>
> We have not shown this directly; however, the experiments in 6.1 may have implicitly verified this, as $\mathcal{N}$-MIENF only works well in cases with Gaussian dependence (see Figs 2c, 2d, 2e).
>
> > **Could the better performance of neural estimators in Sec 6.2 also due to that they do not need to be normalised?**
>
> Yes, particularly considering its deep connection to model flexibility and capacity. However, please don't forget that this flexibility comes at a price (no control over complexity).
>
>
>
> &nbsp;
>
>
> **Other issues**. Issues such as typos and notation issues have been fixed in the revised manuscript. We sincerely thank the reviewer for the very detailed checking and correction.

---

> > ### Comment · Reviewer_A2bS · 2025-08-01
> > **Thank you for the answers**
> >
> > Dear Authors,
> >
> > Thank you for going through my points and providing answers to them. Please could you comment more on W1 and W4 following my points below?
> > - W1 & Q1: The diagnostic provided are convincing, and I trust the authors to include them in the camera-ready.  Would it be possible to provide such diagnostics for other experiments, as they check for a strict requirement of vector ranks?
> > - W2: To avoid confusion on the recommended approach, it might be preferable to remove the comment on joint learning from the main text or replace it with a note saying joint learning is not recommended, according to the analysis in the Appendix.
> > - W3: Thank you for the clarification.
> > - W4: Thank you, I trust the authors to include the relevant discussion for the camera-ready.
> > - W5: In that case, what is the point of Proposition 4, given that Proposition 3 just above it already argues that a VGC can capture the true copula well.
> > - Q2: Thank you for the reply, it is a good argument for using flows here.
> >
> > Best Regards,
> > Reviewer A2bS

---

> ### Author Response · Authors · 2025-08-02
>
> Thank you very much for your prompt response! Please find below our detailed follow-up:
>
> - W1: We promise that **a comprehensive diagnostic will be added to the revised manuscript**, covering diverse patterns, varying dimensionalities and failure cases. The full distribution of the non-diagonal elements in $\Sigma$ will be shown (see the examples under /ablations/vector_ranks_quality in our repo). Tables 3 & 4 below show more examples.
>
> - W2: Thank you for the suggestion! We have replaced the original footnote with the note you proposed, and we now cite the TACtis-2 paper as additional support.
>
> - W4: **All promised modifications will be included in the final version — you have our word**. Please don’t hesitate to share any further relevant literature.
>
> - W5: Good question! While Proposition 3 tells that a VGC is a reasonable approximation up to second order, it does not imply that higher order dependencies can always be ignored. Proposition 4 goes further by suggesting that such high-order dependencies might be negligible for weakly dependent RVs. Our empirical results is consistent with this theory.
>
> Table 3. (Mixture of Triangles). Inspecting the *non-diagonal* elements $\Sigma_{ij}$ in the covariance matrix $\Sigma$ of the estimated vector rank  $\mathbf{\hat{u}}$.
>
> |  | &nbsp; mean of $\Sigma_{ij}$ &nbsp; | &nbsp; 5% quantile of $\Sigma_{ij}$ &nbsp; | &nbsp; 95% quantile of $\Sigma_{ij}$ &nbsp; |
> | --- | --- | --- | --- |
> | d=32 | &nbsp; -0.007 | &nbsp; -0.034 | &nbsp; 0.032 |
> | d=64 | &nbsp; 0.011 | &nbsp; -0.047 | &nbsp; 0.039 |
> |  |  |  |  |
>
> Table 4. (Compressed BERT embeddings). Inspecting the *non-diagonal* elements $\Sigma_{ij}$ in the covariance matrix $\Sigma$  of the vector rank  $\mathbf{\hat{u}}$. The case d=64 is a failure case$^*$.
>
> |  | &nbsp; mean of $\Sigma_{ij}$ &nbsp; | &nbsp; 5% quantile of $\Sigma_{ij}$ &nbsp; | &nbsp; 95% quantile of $\Sigma_{ij}$ &nbsp; |
> | --- | --- | --- | --- |
> | d=16 | &nbsp; 0.010 | &nbsp; -0.048 | &nbsp; 0.075 |
> | d=64 | &nbsp; 0.014 | &nbsp; -0.361 | &nbsp; 0.394 |
> |  |  |  |  |
>
> *Interestingly, we find that even if the vector ranks are not perfectly learned (e.g. d=64 in Table 4 above), our estimator still yields a reasonable estimate. This may be due to that all univariate ranks  $\mathbf{\hat{u}}_d$ are perfectly uniform, so even if  $\mathbf{\hat{u}}_i,  \mathbf{\hat{u}}_j$ occationally exhibit mild dependence, the overall entropy $|H( \mathbf{\hat{u}}_d)|$ remains low, leading to a small bias in Proposition 2.
>
>
>
> We sincerely thank the reviewer again for the constructive feedback and the very enlightening discussion, which have been invaluable in improving the quality of our work.

---

> > ### Comment · Reviewer_A2bS · 2025-08-02
> > **Thank you, I raise my score**
> >
> > Dear Authors,
> >
> > Thank you for the nice discussion. I mistakenly asked you to comment on W4 when what I meant was W5. I believe that your related work discussion is quite complete, especially with the addition of R1, R2 from reviewer 44kJ, so no concerns there.
> >
> > I appreciate the inclusion of more diagnostics on the vector ranks. Having them in the appendix is sufficient I believe. I indeed looked at the full distributions in the repo following your first reply. It looks convincing in showing the ranks are correctly learned for those cases. I find it interesting that your method works well even in the failure case when vector ranks are not quite independent. I believe this could motivate future research, and constitutes a valuable inclusion to the paper!
> >
> > With regards to Proposition 4, I understand your motivation for including it. As it is not incorrect or harmful to the paper I have no further concerns with it.
> >
> > Since all my concerns were adequately resolved, I raise my score to 5. I will happily argue for the acceptance of your paper.

---

> ### Author Response · Authors · 2025-08-02
> **Thank you!**
>
> Please accept our heartfelt gratitude, not only for raising the score to a 5, but also for the very enlightening and rewarding discussion. We have learned a lot from it.
>
> Best regards, Authors

---

### Official Review · Reviewer_DGA1 · 2025-07-03

**Clarity:** 4
**Significance:** 4
**Originality:** 4
**Rating:** 5
**Confidence:** 3

**Summary:**

The paper describes an approach to estimating mutual information that
bridges the gap between neural models and simpler models, e.g.,
Gaussian copula.  For this, they propose to use Vector Copulas.

Experiments show some advantages of the method.

**Questions:**

I would be helpful for the reader if the main text of the paper pointed to the appendix
for the proof of Thm 2  :)

Stronger experimental results would strengthen the work.

**Ethical Concerns:**

["NO or VERY MINOR ethics concerns only"]

**Final Justification:**

I remain positive on this paper.

**Limitations:**

yes

**Paper Formatting Concerns:**

None noted.

**Quality:**

4

**Strengths And Weaknesses:**

MI estimation is an important area, e.g, for optimizing encoders
in representation learning.

The theoretical development is mostly clear.

It would help the reader to define the push-forward notation (the symbol #)

The equivalence of PMI, usually stated as a density ratio,
and the vector copula density seems very interesting, and as the authors show,
it can be used to re-interpret important prior works.

The remainder of the paper demonstrates ways of applying the
resulting ideas to some MI estimation problems, including
modeling the copula as a mixture of copula.

The results in table 1 show some mild improvements in MI estimation
in comparison to other methods, though MRE is competitive.

The results in Figs 1 and 2 are very hard to read, so performance is
hard to evaluate.

I feel that the theoretical contributions outweigh the somewhat mild
experimental result improvements.

---

> ### Author Rebuttal · Authors · 2025-07-29
>
> Thank you for your constructive feedback and the positive assessment! Your recognition that our work provides a novel perspective of PMI, enables important re-interpretion of prior works, and have good theoretical contributions are very encouraging. Below, we address your constructive criticisms regarding **empirical performance** and **presentation**.
>
>
> &nbsp;
>
> ### **Stronger Empirical Performance**
>
> We fully agree on the importance of strong empirical performance. In fact, **our estimator does demonstrate excellent performance across a wide range of settings**, though this is not easy to notice due to readability issues in Figures 1 and 2 (which we have now fixed). Below, we highlight scenarios where our method shines:
>
> Table 1. Scenarios where our method is a clear winner. Cases like A, C and D are particularly challenging and are pain points in existing estimators.
>
> |  | Challenges | Evidence |
> | --- | --- | --- |
> | **A** &nbsp; | High MI | See cases $\rho$=0.8 in Figure 1 and cases $d$=64 in Figure 2 |
> | **B**  &nbsp;| High dimensionality | Figure 2, see cases $d$=64 |
> | **C** &nbsp; | Long-tailed distributions | Figure 1d & 2c (student-t), Figure 1b (exponential) |
> | **D** &nbsp; | Highly-transformed distributions | Figure 2a (Spiral) |
> | **E** &nbsp; | Hetereogenous marginals | Figure 1b |
> | **F** &nbsp; | Non-Gaussian dependencies | Figure 2c (student-t), Figure 2d & 2e (mixtures) |
> |||
>
> (Additionally, our method also gracefully scales to **image** and **text** data, either achieving the best performance or coming very close to the top performer in these cases.)
>
>
>
>
> &nbsp;
>
> ### **Notations & Presentation**
>
> We sincerely thank the reviewer for pointing out the issues with notation and presentation. We have made substantial improvements in the revised version, including:
>
> - (**Figures 1 and 2**) We have make them clean and more readible by (a) increasing the font and figure sizes; (b) avoiding clutter by moving some baseline to the appendix.
> - (**Theoretical proof pointer**) Following your suggestion, we have now pointed the reader to the relevant proof in the appendix for each theoretical result.
> - (**Notations clarity**) We have formally defined the push-forward operation (the symbol “#”) in Section 2 to improve clarity.

---

> > ### Comment · Reviewer_DGA1 · 2025-08-04
> > **enthusiasm raised**
> >
> > Thanks for clearing up the issues in the presentation of the empirical performance.

---

> ### Author Response · Authors · 2025-08-04
> **Thank you!**
>
> Thank you for the prompt acknowledgement! Once again, we sincerely appreciate the time you’ve taken to evaluate our work and your recognition of our work.

---

### Decision · Program_Chairs · 2025-09-17

**Decision:**

Accept (poster)

**Comment:**

The paper shows that mutual information can be recast as a vector copula estimation task. This work generalizes previous results linking copulas and information for univariate variables. A vector copula method for estimation is clearly elaborated and tested, Theoretical arguments motivate the approach, and experiments show its competitiveness compared to existing methods for MI estimation. The paper is relevant because it addresses the very important area of MI estimation. The presented theoretical developments are sound and clear. In particular, Theorem 2 provides a sound direction for future progress. The proposed estimator offers flexibility for selecting an appropriate model capacity for the perceived complexity of a task. These strengths make this contribution relevant to NeurIPS.